# Quantification of the Therapist’s Gentle Pull for Pinch Strength Testing Based on FMA and MMT: An Experimental Study with Healthy Subjects

**DOI:** 10.3390/diagnostics11020225

**Published:** 2021-02-02

**Authors:** Abdallah Alsayed, Raja Kamil, Veronica Rowe, Mazatulfazura S. F. Salim, Hafiz R. Ramli, Azizan As’arry

**Affiliations:** 1Department of Electrical and Electronic Engineering, Faculty of Engineering, Universiti Putra Malaysia, Serdang 43400, Selangor, Malaysia; hrhr@upm.edu.my; 2Laboratory of Computational Statistics and Operations Research, Institute for Mathematical Research, Universiti Putra Malaysia, Serdang 43400, Selangor, Malaysia; 3Department of Occupational Therapy, Georgia State University, Atlanta, GA 30302, USA; vrowe4@gsu.edu; 4Department of Rehabilitation Medicine, Faculty of Medicine and Health Sciences UPM, Serdang 43400, Selangor, Malaysia; fazurasf@upm.edu.my; 5Department of Rehabilitation Medicine, Hospital Pengajar Universiti Putra Malaysia (HPUPM), Serdang 43400, Selangor, Malaysia; 6Department of Mechanical and Manufacturing Engineering, Faculty of Engineering, Universiti Putra Malaysia, Serdang 43400, Selangor, Malaysia; zizan@upm.edu.my

**Keywords:** stroke, FMA, 4/5 MMT, pinch strength, gentle pull, quantitative value

## Abstract

Static pinch strength against a therapist’s gentle pull is evaluated using the pincer grasp component of the Fugl Meyer Assessment (FMA) to assess pinch impairment after stroke. In the pincer grasp component, therapists applied a gentle pull to distinguish between a score of 1 (moderate pinch impairment) and a score of 2 (no pinch impairment). The gentle pull is described as a resistance equivalent to a manual muscle test (MMT) score 4/5. The accepted use of “gentle” as a qualitative description for the pull results is a non-standardized subjective interpretation. The goal of this paper was to determine the quantitative value of the gentle pull applied by the therapists as in their clinical practice using a pinch–pull gripping system. The FMA protocol was used to standardize the body and fingers positions of three occupational therapists who were then instructed to apply a gentle pull of 4/5 MMT using their thumb and index fingers (in a tip-to-tip pinch). The results show that the therapists exerted a mean gentle pull (4/5 MMT score) of 6.34 ± 0.98 N with high reliability and acceptable repeatability. In investigating the ability of healthy subjects to resist the gentle pull, 50 adult male volunteers were instructed to pinch the pincer object and resist a dynamic loading exerted by the pinch–pull gripping system as much as possible to the moment the pincer object slips away. The results show that all subjects were able to exert a pulling force higher than the quantitative value of the gentle pull.

## 1. Introduction

Pinch strength measurement with the wrist in a neutral position is a common evaluation process to determine pinch function after stroke during rehabilitation [1]. It involves measuring the pinch strength at maximal levels as well as pinch strength against an external resistance. The maximal pinch strength is usually evaluated using the Jamar pinch dynamometer, which provides quantitative measurements of pinch force with high reliability and validity [2]. Static pinch strength against external resistance is tested in one component of the Fugl Meyer Assessment (FMA) [3,4]. FMA is the most commonly used assessment for measuring post-stroke impairment [5,6,7]. In the FMA pinch strength test, the patient is required to hold a pencil between the index and thumb fingertips and a therapist applies a gentle pull to the pencil away from the patient’s fingers. The patient’s ability or inability to hold onto the pencil against the gentle pull would lead to either a score of 0 (function cannot be performed—cannot hold pencil), 1 (moderate impairment—can hold pencil, but not against resistance), or 2 (no impairment—can hold the pencil against resistance) [8]. Figure 1 depicts the pinch strength test according to the FMA protocol. In clinical practice, it is difficult to judge the amount of force used during the gentle pull. Therefore, therapists would apply a gentle pull equivalent to a 4/5 score on the manual muscle test (MMT) [9]. However, a 4/5 MMT score is still a qualitative description for the gentle pull, which may cause low intra-rater and inter-rater reliability [10,11]. The use of a quantitative value over a qualitative description of gentle pull would overcome the subjectivity in distinguishing between a score of 1 and a score of 2. 

A therapist’s gentle pull can be measured quantitatively using a data acquisition system. Otten et al. [12] used a hand glove with built-in pressure sensors to acquire the pinch force data of index–thumb finger pinch while applying the gentle pull. Lee et al. [13] overcame the difficulty of wearing the gloves by attaching the pressure sensors to the pincer object directly, and subsequently, the subjects were not required to wear the hand glove. One of the limitations of these systems is related to the performance of the force-sensing resistance (FSR) sensors used to measure the pinch force of the thumb–index fingers, which may jeopardize the accuracy of the pinch force measurements. FSR sensors have low performance in terms of voltage drifting, high hysteresis, and low repeatability [14]. In addition, FSR sensors have some variations in their performance due to the difficulty in applying standard calibration procedures, which results in different calibration outcomes compared to the manufacturer’s datasheet [15]. For instance, in body device applications, human subjects exert inconsistent actuation on the sensing area of the FSR sensor, leading to inconsistent repeatable measurements. Moreover, the variance in the repeatability measurements is due to the pressure distribution of the human fingers being larger than the sensing area [16]. Another limitation is that these systems only measure the axial pinch force exerted by the index–thumb fingers on the pincer object rather than the tangential force which represents the quantitative value of the gentle pull.

Devices, such as handheld dynamometers, are typically used to assess distal muscular strength. However, these devices cannot be used to assess pinch strength against resistance with involvement of the pencil object as in the FMA protocol. Furthermore, they do not include sensors to track the slipping that occurs on the pencil object during the pinch assessment. Therefore, the pinch strength data in this study were collected using a pinch–pull gripping system previously developed [17,18]. Figure 2 depicts the developed system, which mainly consists of a pincer object, linear actuator, pulling force load cell, pinch force load cell, and linear variable differential transformer (LVDT) sensor. The mechanical model of the pincer object was designed and fabricated to mimic the pencil object used in the FMA. The linear actuator was used to pull the pincer object away from the subject’s fingers with dynamic loading. Due to the limitations of the FSR sensors, a customized pinch force load cell was fabricated, such that a strain gauge was fixed on the pincer object to measure the pinch force applied onto the pinching area near the free edge. Subsequently, a Wheatstone bridge and amplifier were used to convert the strain gauge resistance to a voltage signal and amplify the voltage signal, respectively. In measuring the pulling force of subject, a pulling force load cell was placed in between the pincer object and linear actuator. The LVDT displacement sensor was attached to the linear actuator to track the pincer object’s slipping. The performance of the pinch–pull gripping system was mainly determined in terms of linearity, hysteresis, and repeatability. The customized pinch force load cell achieved a high linearity of R^2^ = 0.9886, high repeatability of 1.43 ± 0.447%, and negligible hysteresis of 0.287%. The pulling force load cell achieved a high linearity of R^2^ = 1, high repeatability of 0.01%, and negligible accuracy errors of 0.03%. The displacement sensor achieved high linearity and repeatability of R^2^ = 1 and 9.57 × 10^−5^%, respectively. The pinch–pull gripping system was deemed suitable to collect the pulling force of the therapist’s gentle pull as well as pinch and pulling forces of healthy subjects in this study.

The aim of this study was to determine the quantitative value of the therapist’s gentle pull (4/5 MMT score) using a previously developed pinch–pull gripping system based on FMA and MMT protocols. The following research questions were addressed: (1) What is the quantitative value of a therapist’s gentle pull equivalent to a MMT score of 4/5? (2) Can healthy subjects exert a pulling force higher than the quantitative value of a therapist’s gentle pull? and (3) What is the maximum pulling force that can be resisted by the healthy subjects before the pincer object slips away? Slipping away occurs when the subject is no longer able to exert enough pinch force to keep the pincer object within their fingers. In this study, the recruitment of healthy subjects representing a score of 2 was to test the suitability of the gentle pull. Thus, the authors hypothesized that healthy subjects would exert pulling forces higher than the quantitative value of the gentle pull (4/5 MMT score). In addition, the recruitment of healthy subjects was to determine their reliability in resisting the automatic pull when the pinch–pull gripping system and FMA protocol were used to collect the pinch and pulling force data as well as to standardize their body and fingers positions, respectively. The answer to the first question will provide therapists a new definition for the gentle pull based on a quantitative measurement of gentle pull rather than the qualitative definition of gentle pull based on an MMT score of 4/5. This would tackle the problem related to the subjectivity of using the qualitative description of an MMT score of 4/5. 

## 2. Materials and Methods

### 2.1. Quantification of Gentle Pull

In this study, a single-blind method was adopted in which the therapists applied a gentle pull on the pinch–pull gripping system. The gentle pull was applied in the absence of resistance exerted by the human subjects on the opposite end of the pincer object. The observational experiment to measure the therapist’s gentle pull was conducted at the Universiti Putra Malaysia Teaching Hospital. Three occupational therapists with experience administering the FMA and MMT were recruited. This study was approved on 13 February 2018 by the Ethics Committee of Research Involving Human Subjects of Universiti Putra Malaysia (Reference number: JKEUPM/2017/248). 

In this observational study, the FMA protocol [19,20] was used to standardize the body and fingers positions of the therapists to minimize variability in applying the gentle pull. They were instructed to sit with their hips and knees at 90 degrees of flexion on a chair, with their shoulders in a neutral position, forearm supported on a bedside table, and holding the pincer object using their thumb–index fingers. Consistent with the FMA protocol, the therapists applied some shoulder extension, elbow flexion, and wrist extension to allow a comfortable position while applying the gentle pull. Figure 3 depicts the typical position of each therapist during the observational experiment. Before commencing the experiment, the therapists cleaned their hands to avoid any influence of moisture through sweat. They were also instructed to pinch the pincer object comfortably using their dominant hands and then applied a gentle pull in the direction away from the pincer object with resistance equivalent to a score of 4/5 on the MMT, as practiced in the hospital. Each therapist repeated this procedure for three trials. A video of one trial (Appendix A) is attached to the Appendix A. The duration of one experiment trial started when the therapist pulled the pincer object, indicated by an increase in pulling force data, and ended the instant the therapist released the pincer object, indicated by zero reading of pulling force data. During each trial, one set of pulling force measurements was recorded, resulting in a total of 9 datasets (3 therapists × 3 trials). The pulling force data were incorporated in the reliability test. The intra-rater reliability among the three trials for each therapist was assumed using Cronbach’s alpha as follows:(1)$$\alpha \;=\;\frac{N(v-\bar{c})}{v(N-1)},$$
where *N* is the number of trials, $\bar{c}$ is the average inter-item covariance among the trials, and *v* is the average variance of each trial. The inter-rater reliability among the three therapists was also computed using Cronbach’s alpha, where *N* is the number of therapists, $\bar{c}$ is the average inter-item covariance among the therapists, and *v* is the average variance of each therapist. The peak value of each trial was extracted, which represented the quantitative value of the gentle pull (4/5 MMT score). Subsequently, the intra-rater repeatability indicated the variance of peak values during the three trials of each therapist and it was computed using Equation (2). However, the inter-rater repeatability indicated the variance in the peak values among the three therapists and it was computed as in Equation (3).
(2)$$\mathrm{Intra-rater}\;\mathrm{repeatability}\;=\;\frac{1}{I+1}\sum_{i=1}^{I}\frac{\left|y_{ik}-\bar{y_{k}}\right|}{y_{ik}},$$
where *k* is the index of therapist (*k* = 1, 2, 3), *I* is the total number of trials, $\bar{y_{k}}$ is the mean of peak values obtained from the three trials of therapist *k*, and $y_{ik}$ is the peak value of each trial.
(3)$$\mathrm{Inter-rater}\;\mathrm{repeatability}\;=\;\frac{1}{1+k}\sum_{k=1}^{K}\frac{\left|\bar{y_{k}}-\bar{Y}\right|}{y_{k}},\;\mathrm{and}\;\bar{Y}=\frac{1}{K}\sum_{k=1}^{K}\bar{y_{k}},$$
where *K* is the total number of therapists (*k* = 3), $\bar{Y}$ is the mean of peak values obtained from the three therapists, and $\bar{y_{k}}$ is the mean peak value of each therapist.

### 2.2. Exeprimental Protocol for Healthy Subjects

A single-gender group of fifty right-handed male students participated in this study. All participants passed the health status questionnaire (SF-36) and were free of any arm injury. The study excluded females, left-handed people, professional sportsmen, and construction workers to avoid any bias in pulling force measurements. The subjects had a mean age, height, and weight of 21.86 ± 1.41 years, 170.4 ± 7.03 cm, and 69.49 ± 17.09 kg, respectively. In conducting the observational experiment, the FMA protocol was adapted such that the subject was instructed to sit with their hips and knees at 90 degrees of flexion on a chair, with their forearm rested on a bedside table, elbow at 90 degrees of flexion, and shoulders in a neutral position, as depicted in Figure 4a. The subjects were also instructed to clean and dry their hands before holding the pincer object to avoid any moisture through sweat. Next, each subject was instructed to hold the pincer object of the pinch–pull gripping system comfortably using their index–thumb fingertips, as shown in Figure 4b. Once the linear actuator started to apply the automatic pull on the pincer object in the direction away from the subject, the subject resisted this pull as much as possible. There was no visual feedback provided, hence the subject had to react and adjust his pinching accordingly to the perceived amount of automatic pull. This could be indicated by the proportional increase in pinch force measurements to the changes in the pulling force measurements. The automatic pull was stopped the moment the pincer object had slipped away from the subject’s fingers, and this was indicated by a sudden increase in displacement slip. Note that the automatic pull started from 0 N and increased constantly at a rate of 0.5 N. Right and left hands were measured and each hand performed the experiment for three trials. One trial was recorded from the instant the linear actuator started the automatic pull and ended at the instance of slip-away. The slip-away was defined as the instant the pincer object slipped away from subject’s pinching due to inability to provide enough force to keep the pincer object within his fingers. The analysis of each trial involved the pinch and pulling force values at the slip-away. In total, 300 trials (50 subjects × 2 hands × 3 trials per hand) were recorded such that one trial included pinch force, pulling force, and displacement measurements. Cronbach’s alpha (explained in Equation (1)) was computed to measure the test-retest reliability among the three trials of each hand. In addition, the Pearson product–moment correlation coefficient (*r*) was computed to measure the subject’s ability to exert a pinch force proportional to the pulling force as instructed. Paired *t*-tests were performed on pinch and pulling force values at slip-away to study the significant difference between right and left hands.

## 3. Results

### 3.1. Quantitative Value of Gentle Pull

Figure 5 shows the data recorded from nine trials involving the three therapists while applying the gentle pull equivalent to a 4/5 MMT score. The pulling force signals of the gentle pull were recorded from the pulling force load cell. As the therapists started to pull the pincer object, a sharp increase in the pulling force signal from a relatively flat level was observed. This gives an implication that the subject with a score of 2 should have enough strength response to resist the sharp increase in gentle pull. The peak value of pulling force signal was equivalent to the quantitative value of a 4/5 MMT score. The trial ended the moment the therapist released the pincer object, indicated by an exponential decrease in the pulling force signal towards the zero pulling force level. The peak pulling force values were extracted for analysis. Table 1 summarizes the peak values extracted from the three therapists. The quantitative value of the gentle pull (4/5 MMT score) ranged from 4.32 to 8.31 N and the mean was 6.34 ± 0.98 N. The results suggest that the pulling forces applied by each therapist were consistent, and a mean force of 6.34 N was applied as a gentle pull to distinguish between a score of 1 (moderate impairment) and a score of 2 (no impairment) on the FMA in clinical practice. 

The Cronbach’s alpha values that measured the intra-rater reliability of gentle pull among the three trials were 0.993, 0.985, and 0.979 for each therapist, respectively. In addition, the Cronbach’s alpha value that measured the inter-rater reliability of gentle pull among the three therapists was 0.938. In measuring the variability of the peak value among the three trials for each therapist, the therapists showed intra-rater repeatability values of 9.27%, 7.88%, and 8.15%, respectively. The inter-rater repeatability that measured the variability of peak value among the three therapists was 9.21%.

### 3.2. Pulling Forces of Normal Subjects at Slip-Away

Figure 6 shows the data recorded from three trials of the right hand of a subject. The pinch force, pulling force, and displacement data were collected from the pinch force load cell, pulling force load cell, and LVDT displacement sensor, respectively. The pulling force load cell measured the pulling force of the subject’s pinching against the automatic pull and the force was expected to increase gradually as the automatic pull applied by the linear actuator increased. The linear actuator started to generate the automatic pull indicated by a sudden increase in the pulling force signal at 3 s. The pincer object slipped away from subject’s fingers at 17, 19, and 20 s for the three trials, respectively, as indicated by a sudden decrease in the pulling force signal as well as an increase in the displacement signal. The subject was able to exert a pinch force proportional to the pulling force with Pearson’s (*r*) values of 0.977, 0.846, and 0.950 for the three trials, respectively. During this case, the subject exhibited zero displacement until the slip onset. The subject exhibited little slipping prior to slip-away, especially for the third trial, as seen on the displacement–pulling force curve depicted in Figure 7. The slip-away occurred at 4.01 mm, where the peak pulling force value was recorded. Prior to slip-away, the subject was able to resist the increase in the pulling force, despite the presence of small slip displacement. The Cronbach’s alpha values determining the test-retest reliability among the three trials were 0.99 and 0.969 for pinch and pulling force signals, respectively. In this study, it was important to extract the pinch and pulling force values at slip-away of each trial. Table 2 summarizes these values at slip-away of the trials depicted in Figure 6. 

For each hand of the 50 subjects, average Pearson’s (*r*) values assessed the pinch–pulling force correlation for the three trials to obtain a single value (as demonstrated in Table 2). These single values were then averaged and summarized, as shown in Table 3. In addition, the Cronbach’s alpha values were computed for pinch and pulling signals for each hand. These values were then averaged and are summarized in Table 3 (Full data are provided in Appendix A attached in the Appendix A).

For each hand of the 50 subjects, the pinch and pulling force values at the slip-away of the three trials were averaged to obtain two averaged values (as demonstrated in Table 2). Table 4 summarizes the mean, standard deviation, and range of these averaged values for the right and left hands of the 50 healthy subjects. Full data are provided in Appendix A attached in the Appendix A. The pinch and pulling force values of the subjects’ right hands were higher than those of their left hands by 11.24% and 12.32%, respectively. Based on the paired *t*-test, there were significant differences between the right and the left hands at a confidence level of 0.05 as the *p*-values ≈ 0.

In investigating the ability of healthy subjects to resist the gentle pull, the pulling force of healthy subjects at slip-away was compared with the 6.34 ± 0.98 N gentle pull. As seen in Figure 8, the gentle pull (4/5 MMT score) of therapists was located out of the range (pulling force values of healthy subjects ranged from 7.69 to 25.25 N). Thus, all healthy subjects were able to resist pulling forces higher than the gentle pull (4/5 MMT score). The gentle pull force (6.34 N) was weaker than the mean pulling force of the right (14.84 N) and left (13 N) hands of the volunteers by 57.27% and 51.23%, respectively. In addition, the weakest hand exhibited a pulling force higher than the gentle pull by 21.29%. 

## 4. Discussion

In this study, the quantitative value of the therapist’s gentle pull was 6.34 N. The results showed that the quantitative value varied from 4.32 to 8.31 N among therapists, which was expected as each therapist had a different physical strength and clinical experience in administration of the FMA and MMT. Moreover, there was no visual feedback provided to the therapists, and they had to judge the 4/5 MMT score as in clinical practice. Despite that, each therapist showed high intra-rater repeatability in exerting the peak value (equivalent to 4/5 MMT score) among the three trials, as the repeatability values were less than 10% [21]. The three therapists showed a high inter-rater repeatability value of 9.21%. In measuring the reliability of the therapists in applying the gentle pull over the trial period, each therapist showed high intra-rater reliability among the three trials as the Cronbach’s alpha values were more than 0.9 [22]. The inter-rater reliability was also high as the Cronbach’s alpha values was 0.938. This indicated that standardization of the therapist’s posture based on the FMA protocol reduced the potential subjectivity in applying the gentle pull.

The main factor leading to variability in applying a reliable gentle pull was attributed to the different interpretations of using the FMA protocol by therapists [20]. The difference in interpretations involved the amount of gentle pull as well as the posture of therapists while applying the gentle pull. In this study, the amount of gentle pull, which evaluates the strength of thumb–index fingers, was exerted with some resistance equivalent to a 4/5 MMT score. The 4/5 MMT score was described to the therapists as the resistance that prompted the subject to provide enough effort to resist, but not to break the subject’s ability to resist the gentle pull. However, a therapist may exert different amounts of gentle pull (4/5 MMT score) due to different characteristics of subjects, such as age, body size, and health status [23]. In addition, the subjects may exert extra resistance in the opposite direction of gentle pull (4/5 MMT score), resulting in extra variations in the quantitative value of gentle pull. Therefore, in quantifying the gentle pull, the therapists were instructed to apply the gentle pull (4/5 MMT score) against a fixed end, represented by the linear actuator of the pinch–pull gripping system. This method guaranteed that the quantitative value measured using the pulling force load cell was only equivalent to the amount of gentle pull (4/5 MMT score) applied by therapists. Note that the amount of gentle pull applied by Asian therapists can be different to that of European and African therapists, as the MMT score of 4/5 would be influenced by therapist strength and testing technique [24]. In minimizing the variability resulting from therapist’s posture, their shoulder, elbow, and forearm were in the same position as described in Section 3.1 for each trial. This allowed the therapist to judge the amount of gentle pull (4/5 MMT score) he/she exerted by the receptors of thumb–index fingers, but not by the shoulder, elbow, and wrist proprioceptors. However, it was acceptable for therapists to apply some shoulder and wrist extension and elbow flexion to guarantee a comfortable ability to apply the gentle pull. In addition, a bedside table was provided to support for their arm against gravity resistance.

According to the FMA, the pincer object slips away due to the inability of the subject to resist the pulling force of the therapist’s gentle pull. In this study, it was hypothesized that the maximum pinch and pulling forces of healthy subjects would be at the initial moment of slip-away just before the point where the slip displacement suddenly increased. Interestingly, it was found that all healthy subjects exhibited initial slip (slip onset) much earlier than slip-away. However, the slip displacement that occurred prior to slip-away was small (up to few millimeters) and the subjects were still able to exert higher pinch forces as the pulling force increased. Hence, in clinical practice, the therapist should expect a small slip exhibited by the patient and a score of 1 should be given only in the case that the pincer object completely slips away from patient’s fingers. Since the pinching location was at the free end of the pincer object, subjects released the pincer object at slip-away probably due to the overwhelming pulling force generated by the linear actuator or because there was not enough contact surface between their fingertips and pincer object. The results showed that the pinch and pulling force values at slip-away varied among the healthy subjects, which was expected, as each subject has different body and hand characteristics. For example, a bigger fingertip size would lead to a larger contact area between the fingertip and the pincer object’s surface. Consequently, a larger contact area may lead to larger pinch and pulling forces, as reported in a previous study by Derler et al. [25]. Moreover, skin characteristics, such as moisture, roughness, and oiliness, of each subject may influence the friction between their fingertip skin and the object’s surface [26]. The results showed that the healthy subjects exerted pulling forces at the slip away much higher than the pulling force equivalent to the gentle pull. This indicated that the quantitative value of the therapist’s gentle pull was suitable to be used as a threshold to distinguish healthy subjects representing a score of 2. Furthermore, this would open the possibility to track the improvement of pinch impairment beyond the quantitative value of gentle pull, such that patients with stroke can exert a pulling force higher than the gentle pull over rehabilitation time. The continuous improvement in pulling force would improve the responsiveness of the pincer grasp assessment component. In addition, tracking the improvement of pinch impairment to the moment of slip away can solve the problem related to the ceiling effect of the pincer grasp component of FMA, in which the patients with a score of 2 can still exhibit stronger pulling force over rehabilitation time.

This study demonstrates that high test-retest reliability in the right and left hands of healthy subjects was obtained as the pinch–pull gripping system and FMA protocol were used to collect pinch and pulling force data as well as to standardize their body and fingers positions. Therefore, the pinch–pull gripping system can be used to provide accurate measurements to assist therapists in the interpretation of results in order to accurately draw conclusions with minimal effect of external factors. The results of this study also showed that healthy subjects were able to adjust their pinch force in proportion to the pulling force, as indicated by the strong linear association (Pearson’s (*r*) > 0.8) between pinch and pulling forces during the trials for right and left hands. However, there was some inconsistency in the Pearson’s (*r*) values among the subjects which may be attributed to the safety margin, which is an additional amount of force exerted to guard against slip [27,28]. Since the subjects had to adapt their pinch force to the perceived amount of automatic pull without visual feedback, the safety margin was continuously changing to adapt to the continuous increase in pulling force. Hence, it was possible that some subjects may experience unconscious failure to adapt their pinch force to the unpredictable increase in pulling force during the trial. 

In investigating the ability of healthy subjects to resist the automatic pull, the results showed that all healthy subjects were able to resist a pulling force higher than the quantitative value of the therapist’s gentle pull prior to slip-away. These results were limited to the population recruited in this study and cannot be generalized for subjects of other genders and age groups. This population was a single-gender group and was selected to ensure the recruitment of a large-enough number of subjects to investigate the pulling force in a homogenous population. Recruiting subjects from other genders and age groups may show an overlap between gentle pull and pulling force data of healthy subjects. This overlap would be expected, as the physical ability of females and older people is lower than the adult male students recruited in this study [29,30]. Thus, it is advantageous to test the quantitative value of the therapist’s gentle pull among females and people of different ages to draw more comprehensive conclusions. This study utilized healthy subjects to establish a proof of concept and that future studies should involve stroke survivors to test the quantitative gentle pull among those with a score of 1 on the FMA. In addition, the pinch–pull gripping system can be used to apply a gentle pull with a force equivalent to the quantitative value of a therapist’s gentle pull against a patient’s pinching ability on the pincer object. In this study, the threshold value that distinguished between a score of 1 and a score of 2 on the FMA was determined based on therapist’s gentle pull. Further studies should be conducted to determine the threshold based on a data classification method. In this method, the threshold can be defined as the pulling force value that distinguished between the pulling force data of healthy subjects (representing a score of 2 on the FMA) and patients with stroke (representing a score of 1 on the FMA).

## 5. Conclusions

In this study, the therapist’s gentle pull, qualitatively described as an MMT score of 4/5, was quantitatively determined. The therapists applied 6.34 ± 0.98 N as a gentle pull (4/5 MMT score) to distinguish between a score of 1 and a score of 2 on the FMA. The results showed that all therapists exhibited high reliability and acceptable repeatability in applying the gentle pull. Fifty healthy subjects were able to resist a pulling force higher than the quantitative value of the therapist’s gentle pull. The gentle pull value of 6.34 N was lower than the mean pulling forces of the right (14.84 N) and left (13 N) hands of the subjects by 57.27% and 51.23%, respectively. In addition, the healthy subjects exhibited high test-retest reliability in exerting pinch and pulling forces against the automatic pull applied by the pinch–pull gripping system. A further study can be conducted to administer the gentle pull automatically using the pinch–pull gripping system, involving patients with stroke. 

## Figures and Tables

**Figure 1 diagnostics-11-00225-f001:**
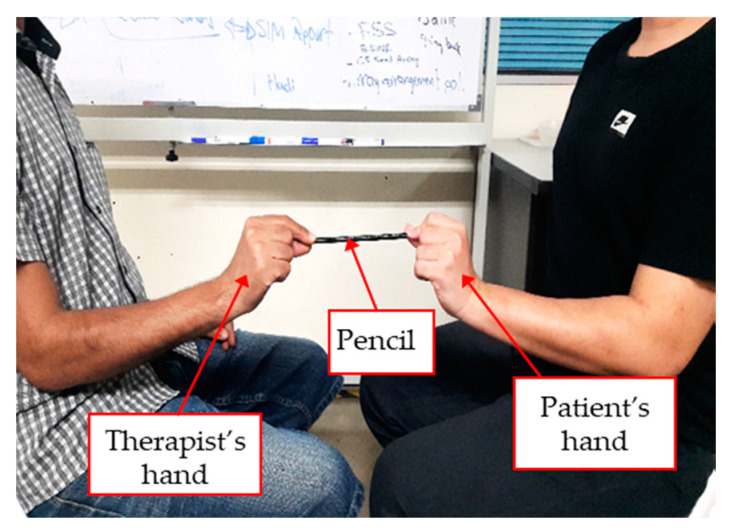
Pinch test setup according to the Fugl Meyer Assessment (FMA).

**Figure 2 diagnostics-11-00225-f002:**
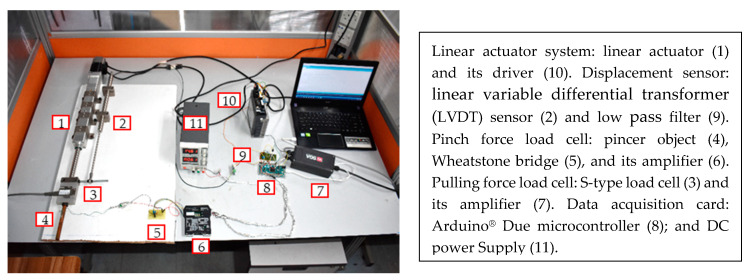
Experimental setup of the pinch–pull gripping system.

**Figure 3 diagnostics-11-00225-f003:**
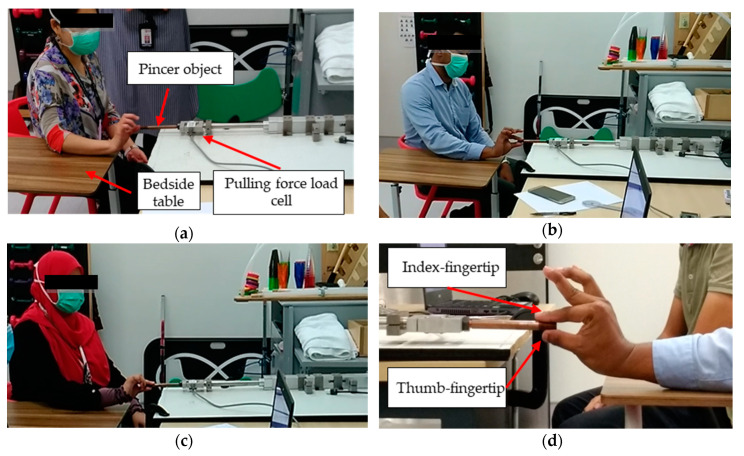
(**a**–**c**) Body postures of the therapists and (**d**) thumb–index fingertips opposed onto the pincer object.

**Figure 4 diagnostics-11-00225-f004:**
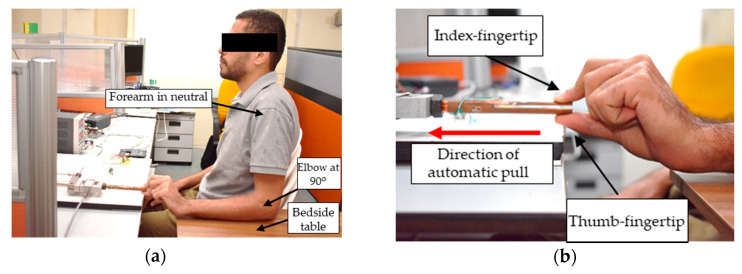
(**a**) Body posture during the experiment; and (**b**) thumb–index finger opposed onto the free end of the pincer object.

**Figure 5 diagnostics-11-00225-f005:**
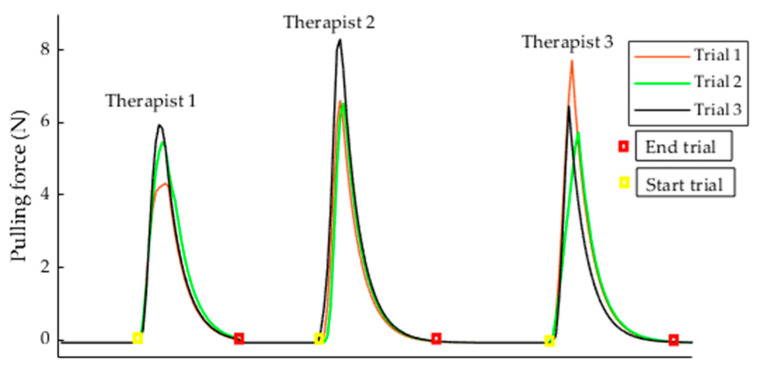
Nine trials of gentle pull collected from therapists.

**Figure 6 diagnostics-11-00225-f006:**
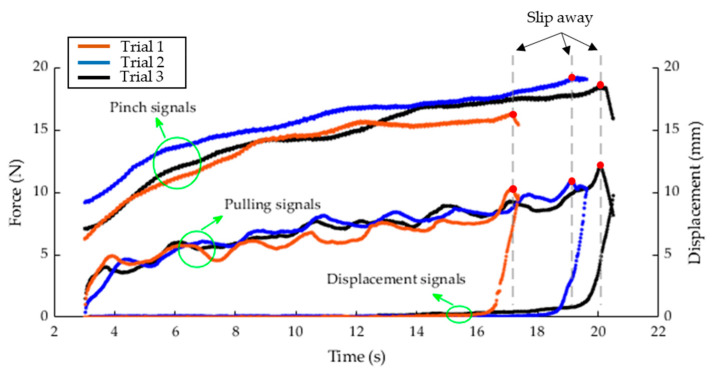
Three trials collected from the right hand of a healthy subject. Red dots (●) represent the pinch and pulling force values at the slip-away.

**Figure 7 diagnostics-11-00225-f007:**
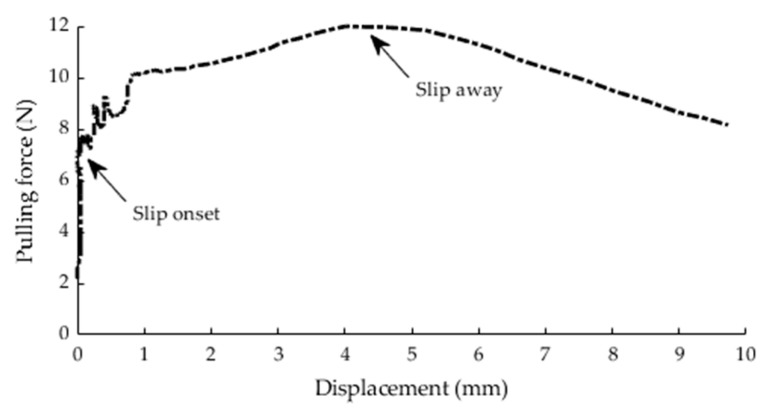
Displacement–pulling force curve of third trial.

**Figure 8 diagnostics-11-00225-f008:**
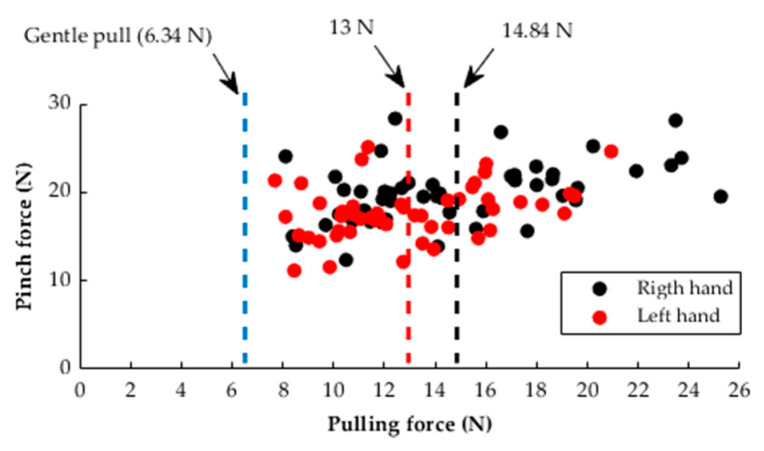
Distribution of pulling–pinch force values at slip-away for 50 subjects compared to the quantitative value of gentle pull.

**Table 1 diagnostics-11-00225-t001:** Peak values (N) representing the quantitative value of a 4/5 manual muscle test (MMT) score.

Trial	Therapist 1	Therapist 2	Therapist 3	
1	4.32	6.61	7.73	
2	5.46	6.54	5.74	
3	5.95	8.31	6.46	
Mean	5.24	7.15	6.64	6.34

**Table 2 diagnostics-11-00225-t002:** Pinch and pulling force values at slip-away of three trials in Figure 6.

Variable	Trial 1	Trial 2	Title 3	Mean
Pinch force (N)	16.3	19.21	18.43	17.98
Pulling force (N)	10.27	10.94	11.98	11.06
Pinch-pulling Pearson’s (*r*)	0.977	0.846	0.950	0.924

**Table 3 diagnostics-11-00225-t003:** Cronbach’s alpha and Pearson’s (*r*) values for 50 healthy subjects.

Variable	Cronbach’s Alpha		Pearson’s (*r*)	
	R	L	R	L
Pinch force	0.918 ± 0.05	0.907 ± 0.05	0.917 ± 0.03	0.893 ± 0.05
Pulling force	0.899 ± 0.05	0.908 ± 0.04		

R: Right hands; L: Left hands.

**Table 4 diagnostics-11-00225-t004:** Summary of pinch and pulling force values at slip-away of 50 subjects.

Variable		Mean	Standard Deviation	Range
Pinch force (N)	Right hand	19.92	3.48	12.35–28.34
	Left hand	17.68	3.09	11.15–25.12
Pulling force (N)	Right hand	14.84	3.57	8.11–25.25
	Left hand	13	2.72	7.69–20.93

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
