# Peer review of "Quantification of the Therapist’s Gentle Pull for Pinch Strength Testing Based on FMA and MMT: An Experimental Study with Healthy Subjects"

_diagnostics, 2021, doi:10.3390/diagnostics11020225_

Round 1

Reviewer 1 Report

This work shows the development of a standardized approch to the evaluation of muscular stenght. Alghout interesting the paper the lack of focus of what is this work about. Methods are well described Hovewever, - what could the be the importance of having a quantitative test in clinical practive ou investigation. I don't forsee this in might everyay practice. - Introdution is too long. Must be resumed and incorporate also related work. - Why have not studied the other - How does this tell us that grades - It sees to be a preleminary despriction and does not actually prove to be reliable in disease because this was not tested anyway - From the clinical point of view we need to see how does it work in patients with motor paresis. Otherwise this is just a physiological test

Author Response

We would like to thank the reviewer for the comments and feedback. Please see the attachment.

Reviewer 2 Report

The paper presents a study regarding a static pinch strength evaluation against therapist gentle pull. The paper is very well written and the methodology is well presented.

There are a few aspects there are not very clear.

First of all, what is the final goal of the paper? The authors have indicated the following issues:

“1) What is the quantitative value of therapist’s gentle pull equivalent to a MMT score of 4/5? 2) Can healthy subjects resist the quantitative value of a therapist’s gentle pull? and 3) What is the maximum pulling force that can be resisted by the healthy  subjects before the pincer object slips away?”

So why question no. 1? Why is it important for patient treatment or his state assessment? Couldn’t you use a regular device for this task?

Regarding question 2: isn’t this eligible for experimental study?

Question no. 3: why is it important in real life?

Do the authors propose a method in which the therapist should learn how to evaluate correctly the patient’s pincer strength?

What are the advantages of the developed device compared to commercially existing ones?

The study itself is not very meaningful since only healthy subjects have been used (and only males).

Author Response

We would like to thank the reviewer for the comments and feedback.

Point 1: First of all, what is the final goal of the paper?

Response 1: In clinics, therapists use the gentle pull to distinguish between patients with pinch impairment (score 0 or 1) and those with no pinch impairment (score 2). The accepted use of “gentle” as a qualitative description for the pull results in a non-standardized subjective interpretation. Therefore, the goal of this paper was to determine the quantitative value of the gentle pull applied by the therapists as in their clinical practice using a pinch-pull gripping system. In addition, the paper investigates the reliability and repeatability of therapists to apply the gentle pull when the FMA protocol was used to standardize their body and fingers positions. The abstract has been revised to clearly explain the goal of the paper.

Original abstract:

The static pinch strength against therapist’s gentle pull is evaluated using the pincer grasp component of Fugl Meyer Assessment (FMA) to determine the level of pinch impairment after stroke. In the pincer grasp component, the therapist applies a gentle pull with a resistance equivalent to a Manual Muscle Test (MMT) score of 4/5. However, the qualitative description is subjective which may result in low inter-rater and intra-rater reliability. In this study, the quantitative value of the therapist’s gentle pull is determined. Three occupational therapists applied a gentle pull of 4/5 MMT using their thumb and index fingers (in a tip-to-tip pinch). A pinch-pull gripping system has been developed to measure the pulling force exerted during the gentle pull. The results show that the therapists exerted a mean gentle pull (4/5 MMT) of 6.34±0.98 N with high reliability and acceptable repeatability. In investigating the ability of healthy subjects to resist the gentle pull, 50 adult male volunteers were instructed to pinch the pincer object and resist a dynamic loading exerted by the pinch-pull gripping system as much as possible to the moment the pincer object slips away. The results show that all subjects were able to exert pulling force higher than the quantitative value of the gentle pull.

Revised abstract:

The static pinch strength against therapist’s gentle pull is evaluated using the pincer grasp component of Fugl Meyer Assessment (FMA) to assess the pinch impairment after stroke. In the pincer grasp component, the therapists applied a gentle pull to distinguish between a score of 1 (moderate pinch impairment) and a score of 2 (no pinch impairment). The gentle pull is described as a resistance equivalent to a Manual Muscle Test (MMT) score 4/5. The accepted use of “gentle” as a qualitative description for the pull results in a non-standardized subjective interpretation. The goal of this paper was to determine the quantitative value of the gentle pull applied by the therapists as in their clinical practice using a pinch-pull gripping system. The FMA protocol was used to standardize the body and fingers positions of three occupational therapists who are then instructed to apply a gentle pull of 4/5 MMT using their thumb and index fingers (in a tip-to-tip pinch). The results show that the therapists exerted a mean gentle pull (4/5 MMT) of 6.34±0.98 N with high reliability and acceptable repeatability. In investigating the ability of healthy subjects to resist the gentle pull, 50 adult male volunteers were instructed to pinch the pincer object and resist a dynamic loading exerted by the pinch-pull gripping system as much as possible to the moment the pincer object slips away. The results show that all subjects were able to exert pulling force higher than the quantitative value of the gentle pull.

Point 2: So why question no. 1? Why is it important for patient treatment or his state assessment?

Response 2: The answer to this question will provide the therapists a new definition for the gentle pull such that the gentle pull is defined quantitatively rather than the description of a MMT score of 4/5. This would tackle the problem of subjectivity of using the qualitative description of a MMT score of 4/5. The authors provide explanation on the importance of question 1 in the Introduction.

Revised: Line 113

The answer to the first question would provide the therapists a new definition for the gentle pull based on the quantitative measurement of gentle pull rather than the qualitative definition of gentle pull based on a MMT score of 4/5. This would tackle the problem related to the subjectivity of using the qualitative description of a MMT score of 4/5.

Point 3: Couldn’t you use a regular device for this task?

Response 3: Devices, such as handheld dynamometers, are typically used to assess strength of distal muscular strength. However, these devices cannot be used to assess the pinch strength against resistance with involvement of the pencil object as in the FMA protocol. Furthermore, they do not include sensors to track the slipping that occurs to the pencil object during the pinch assessment. In literature, there is no device specific for pinch test of FMA. Thus, in this study, the device was developed to meet the requirements of FMA protocol such that the mechanical model of the pincer object was designed and fabricated to mimic the pencil object used in the FMA. The load cell was connected to the pincer object in order to measure the pulling force of therapist’s gentle pull. This explanation is added to the Introduction in the revised paper (previously Related Work).

Revised: Line 75

Devices, such as handheld dynamometers, are typically used to assess strength of distal muscular strength. However, these devices cannot be used to assess the pinch strength against resistance with involvement of the pencil object as in the FMA protocol. Furthermore, they do not include sensors to track the slipping that occurs to the pencil object during the pinch assessment. Therefore, the pinch strength data in this study was collected using a pinch-pull gripping system previously developed [17,18]. Figure 2 depicts the developed system which mainly consists of a pincer object, linear actuator, pulling force load cell, pinch force load cell, and Linear Variable Differential Transformer (LVDT) sensor. The mechanical model of the pincer object was designed and fabricated to mimic the pencil object used in the FMA. The linear actuator was used to pull the pincer object away from the subject’s fingers with dynamic loading.  

Point 4: Regarding question 2: isn’t this eligible for experimental study?

Response 4: We don’t understand exactly what the reviewer means by this. We are thinking maybe the reviewer sees the question as originally written to not be answered by the current study. Thus, Question 2 has been restated.

Original:

2) Can healthy subjects resist the quantitative value of a therapist’s gentle pull?

Revised:

2) Can healthy subjects exert pulling force higher than the quantitative value of a therapist’s gentle pull?

Point 5: Question no. 3: why is it important in real life?

Response 5: In the case of the FMA (pinch strength against gentle pull), it is unknown whether healthy subjects exert his/her maximum pulling force against the therapist’s gentle pull or not. The results show that the healthy subjects exerted the maximum pulling force at slip away much higher than the pulling force equivalent to the gentle pull. This would open the possibility to track the improvement of pinch impairment beyond the quantitative value of the gentle pull. In other words, the patient can exert higher pulling force than the gentle pull over rehabilitation time. In addition, the continuous improvement on pulling force can improve the responsiveness of the FMA. On the other hand, the quantitative measurement can tackle the problem related to the ceiling effect of FMA such that patients with a score of 2 may still have pinch impairment. Thus, tracking the pinch improvement at pulling force higher than the quantitative value of gentle pull would provide more accurate assessment. The authors add this explanation to the Discussion.

Revised: Line 314-324

The results showed that the healthy subjects exerted pulling force at the slip away much higher than the pulling force equivalent to the gentle pull. This indicated that the quantitative value of therapist’s gentle pull was suitable to be used as a threshold to distinguish healthy subjects representing a score of 2. Furthermore, this would open the possibility to track the improvement of pinch impairment beyond the quantitative value of gentle pull such that patients with stroke can exert pulling force higher than the gentle pull over rehabilitation time. The continuous improvement on pulling force would improve the responsiveness of the pincer grasp assessment component. In addition, tracking the improvement of pinch impairment to the moment of slip away can solve the problem related to the ceiling effect of pincer grasp component of FMA in which the patients with a score of 2 can still exhibit stronger pulling force over rehabilitation time.

Point 6: Do the authors propose a method in which the therapist should learn how to evaluate correctly the patient’s pincer strength?

Response 6:

In this study, the FMA guideline improved by Sullivian et al. [16] and Page et al. [17] were adopted. However, these protocols are only limited to standardize the patient’s posture during the assessment. In this study, we used the same protocols to standardize the therapist’s posture as well. This would minimize the variability among the therapists to exert a gentle pull equivalent to 4/5 MMT. This explanation is added to the subsection 2.1 Quantification of gentle pull.

Revised: Line 128

In this observational study, the FMA protocol [16, 17] was used to standardize the body and fingers positions of the therapists to minimize variability in applying the gentle pull.

Point 7: What are the advantages of the developed device compared to commercially existing ones?

Response 7:

As mentioned in Point 3, the regular devices cannot be used to assess the pinch strength with involvement the pencil object as in the FMA. In addition, these devices cannot detect the moment of slip away at which the pincer object totally slipped. These devices are only limited to measure the resistance applied by the patient when the postural muscles are tested. In this study, the developed device mainly included the pincer object, load cell, linear actuator, and LVDT displacement sensor. The mechanical model of the pincer object was developed to mimic the pencil object used in the FMA. The load cell was connected in series to the pincer object to measure the pulling force generated when the therapist applied the gentle pull. The LVDT sensor was implemented to detect the moment of slip away. The linear actuator was used to apply automatic pull with constant force rate to the moment of slip away. In the future work, the linear actuator can be used to apply the gentle pull with a force equivalent to the quantitative value of therapist’s gentle pull which would minimize the subjectivity in applying the gentle pull. The advantages of the developed device are added in the Introduction (Related work in the original paper). The future use of the developed device is added in the Discussion.

Revised: Line 75

Devices, such as handheld dynamometers, are typically used to assess strength of distal muscular strength. However, these devices cannot be used to assess the pinch strength against resistance with involvement of the pencil object as in the FMA protocol. Furthermore, they do not include sensors to track the slipping that occurs to the pencil object during the pinch assessment. Therefore, the pinch strength data in this study was collected using a pinch-pull gripping system previously developed [17,18]. Figure 2 depicts the developed system which mainly consists of a pincer object, linear actuator, pulling force load cell, pinch force load cell, and Linear Variable Differential Transformer (LVDT) sensor. The mechanical model of the pincer object was designed and fabricated to mimic the pencil object used in the FMA. The linear actuator was used to pull the pincer object away from the subject’s fingers with dynamic loading.

Revised: Line 351

In addition, the pinch-pull gripping system can be used to apply the gentle pull with a force equivalent to the quantitative value of therapist’s gentle pull against a patient’s pinching ability on the pincer object.

Point 8: The study itself is not very meaningful since only healthy subjects have been used (and only males).

Response 8: The quantitative value of gentle pull can be tested among subjects with a score 1 and those patients with a score 2 on the FMA. This study utilized healthy subjects to establish a proof of concept and that future studies would involve stroke survivors with a score of 1. This limitation is added to the revised paper.

Revised: Line 349

This study utilized healthy subjects to establish a proof of concept and that future studies would involve stroke survivors to test the quantitative gentle pull among those with a score of 1 on the FMA.

Round 2

Reviewer 1 Report

The title needs to state "healthy subjects" to be in line with the study aims and conclusions

Author Response

Thank you so much for improving the paper. 

The title has been revised to be:

"Quantification of Therapist’s Gentle Pull for Pinch Strength Testing Based on FMA and MMT: An Experimental Study with Healthy Subjects"

Reviewer 2 Report

The authors have successfully adressed the issues mentioned in the first review. 

Author Response

We appreciate your time and comments that improve the paper.